# Potential of eye-tracking simulation software for analyzing landscape preferences

Uta Schirpke[1,2‡]*, Erich Tasser[2‡], Alexandros A. Lavdas[3,4]

**1** Department of Ecology, University of Innsbruck, Innsbruck, Austria, **2** Institute for Alpine Environment, Eurac Research, Bozen/Bolzano, Italy, **3** Institute for Biomedicine, Affiliated Institute of the University of Lübeck, Eurac Research, Bozen/Bolzano, Italy, **4** The Human Architecture & Planning Institute, Inc., Concord, MA, United States of America

‡ US and ET are joint first authors on this work.
* Uta.Schirpke@uibk.ac.at

**Data Availability Statement:** All relevant data are within the paper and its Supporting Information files.

**Funding:** This work was supported by the Department of Innovation, Research, University and Museums of the Autonomous Province of

## Abstract

Profound knowledge about landscape preferences is of high importance to support decision-making, in particular, in the context of emerging socio-economic developments to foster a sustainable spatial development and the maintenance of attractive landscapes. Eye-tracking experiments are increasingly used to examine how respondents observe landscapes, but such studies are very time-consuming and costly. For the first time, this study explored the potential of using eye-tracking simulation software in a mountain landscape by (1) identifying the type of information that can be obtained through eye-tracking simulation and (2) examining how this information contributes to the explanation of landscape preferences. Based on 78 panoramic landscape photographs, representing major landscape types of the Central European Alps, this study collected 19 indicators describing the characteristics of the hotspots that were identified by the Visual Attention Software by 3M (3M-VAS). Indicators included quantitative and spatial information (e.g., number of hotspots, probabilities of initially viewing the hotspots) as well variables indicating natural and artificial features within the hotspots (e.g., clouds, lighting conditions, natural and anthropogenic features). In addition, we estimated 18 variables describing the photo content and calculated 12 landscape metrics to quantify spatial patterns. Our results indicate that on average 3.3 hotspots were identified per photograph, mostly containing single trees and tree trunks, buildings and horizon transitions. Using backward stepwise linear regression models, the hotspot indicators increased the model explanatory power by 24%. Thus, our findings indicate that the analysis of eye-tracking hotspots can support the identification of important elements and areas of a landscape, but it is limited in explaining preferences across different landscape types. Future research should therefore focus on specific landscape characteristics such as complexity, structure or visual appearance of specific elements to increase the depth of information obtained from eye-tracking simulation software.

Bozen/Bolzano. The authors thank the Department of Innovation, Research, University and Museums of the Autonomous Province of Bozen/Bolzano for covering the Open Access publication costs. The funders had no role in study design, data collection and analysis, decision to publish, or preparation of the manuscript.

**Competing interests:** The authors have declared that no competing interests exist.

## Introduction

Appealing landscapes are important for physical and mental well-being [1], providing not only aesthetic values but also recreational spaces [2]. In particular, mountain landscapes are highly appreciated by residents and visitors due to their high degree of naturalness, landscape diversity and long vistas, originating from the complex topography, climatic variability and traditional agricultural use of the landscape [3–6]. Aesthetic quality of such landscapes also generates economic benefits and pictures of beautiful landscapes are often used for marketing purposes [7, 8]. However, ongoing landscape changes driven by global change pressures, mostly resulting in altered agricultural practices [9, 10], also lead to changes in spatial landscape patterns and mountain scenery [11]. For example, the abandonment of mountain pastures leads to an increase in forest due to natural forest regrowth [12], which restricts the viewing depth, reduces landscape diversity and result in less attractive mountain landscapes [13, 14]. In contrast, the valley bottoms are mostly affected by the intensification of agricultural use as well as the expansion of settlements and infrastructure [14, 15], increasing the share of less preferred artificial features and landscape types [16–18], which also induces a decline in aesthetic landscape values [11, 14]. As such developments are ongoing due to legacy effects [19, 20] and expected future demand for ecosystem services [14], profound knowledge about landscape preferences is of high importance to support decision-making in developing effective strategies for maintaining preferred visual landscape characteristics and fostering positive benefits for human well-being [11, 21].

Research on landscape perceptions has been evolved over long time [22, 23]. To gather people's preferences, stated-preference approaches such as photo-based surveys or on-site interviews are widely applied [4, 24–26]. In particular, photo rating is a standard method that can capture perceptions of individuals or groups in a reliable way [27]. For this purpose, panoramic photographs are considered useful stimuli to holistically depict the surrounding landscape [28, 29], as they are easier to recognize and memorize and therefore the responses are more adequate and detailed than for standard photographs [30]. To understand and map aesthetic landscape values, researchers have related landscape preferences to visual landscape characteristics and spatial patterns [24, 25, 31–33]. Such studies revealed, for example, that mean levels of diversity and complexity are generally preferred over highly complex or homogeneous patterns [26, 33, 34] or that landscape openness is positively related to landscape preferences, which can be measured through visibility metrics [25]. There is also a growing body of literature using novel approaches and new data sources such as social media data to identify landscape preferences and the level of aesthetic value [16, 18, 28, 35, 36].

However, all these approaches do not reveal how respondents observe the landscapes, i.e., at which parts they look, how long and in which sequence. Such information can be obtained in an objective way by eye-tracking experiments, as this technique measures the position and number of fixations, the fixation duration, the number of saccades and their direction and velocity as well as the observed horizontal area and the observed vertical area [30, 37, 38]. Although eye-tracking techniques have been used in psychology, marketing, geography, cartography and landscape planning already since long time [37, 39–43], they have been applied in landscape perception research only more recently. For example, eye-tracking experiments were used to examine landscape perceptions in relation with the level of openness and heterogeneity in rural landscapes [30] as well as different levels of urbanization and visual complexity along a rural-urban gradient [44]. Other studies focused on the beauty of natural environments, e.g., forests [38, 45–47], underwater reefs [48] or lakescapes [49] as well as visual preferences in urban environments and urban green spaces, e.g., [33, 50–55]. Eye-tracking experiments can also support the understanding individual's perceptions and cognitive

processes for decision-making related to landscape and ecosystem service planning as shown for mountain landscapes [21] and forest landscapes [47]. In summary, eye-tracking is a well-situated approach in context with landscape perception without the language filter [55].

A great disadvantage of real eye-tracking experiments involving individuals is that they are very time-consuming and costly; studies are therefore often limited by a small sample size and a convenience sample [50]. To overcome such issues, artificial intelligence applications simulating initial eye-tracking movements on a picture are increasingly used, for example, to test the visibility of specific elements such as street signs or crosswalks to increase traffic safety [56–58]. Hollander et al. [54] compared the outcomes between laboratory-based eye-tracking experiments and those generated by simulation software in the context of urban design perceptions; the results were highly consistent suggesting that simulation results can be used as a proxy for laboratory-based eye-tracking results. Accordingly, first studies used eye-tracking simulation software in urban settings, focusing on streetscapes and facades, providing promising perspectives for urban planning [59–61]. These studies used Visual Attention Software by 3M (3M-VAS), a proprietary artificial intelligence software, which simulates "first glance vision". Eye-tracking simulation of the initial fixations can provide information on the pre-attentive processing of visual components that has taken place [59]. The visual system can select salient information to guide appropriate responses with survival value. This process initiates in the retina, where computation of low-level visual features is started, and continues in the thalamus and the early visual cortical areas [62]. The retina is thereby not only a photoreceptor array, but the horizontal interconnectivity between retinal cells also allows a first level of processing [63]. Neurons at these initial perception level are tuned to react to simple visual properties, e.g., color opponency, intensity contrast as well as, in the visual cortex, orientation, direction and velocity of motion, etc. [64]. These visual features are calculated pre-attentively in a parallel manner producing an initial "saliency map" [65]. Information from the initial, pre-attentive processing of visual input, which lasts approximately 200–250 ms, is then used to guide the early deployment of selective attention [66].

However, this technology is not yet sufficiently explored for rural and (semi-)natural environments such as mountain landscapes, as it has only been tested in urban settings [59–61]. It remains unclear, which information that is generated by eye-tracking simulation software can be useful for explaining landscape preferences and whether this information can enhance modelling approaches used to estimate aesthetic landscape values [11]. To address this gap, this study explores the potential of integrating eye-tracking simulation software into research on landscape preferences, focusing on (1) the type of information that can be obtained through eye-tracking simulation in mountain landscapes, and (2) how this information can contribute to the explanation of landscape preferences. Based on previous studies that gathered landscape preferences using panoramic landscape photographs through surveys as well as studies linking such preferences to landscape characteristics, this study analyses and integrates for the first time initial eye-tracking movements generated by 3M-VAS.

## Materials and methods

### Methodological approach

In the context of previous research on visual perception and visual decision-making [67], most studies focused on the consumer and his/her visual attend to advertisements, packaging of products or label design [68], whereas the research object of this study is the landscape. In general, attention and interest to visual stimuli are triggered by two groups of factors, which are top-down and bottom-up attention factors [68]. Top-down factors, or endogenous and goal-directed attention, were often the focus of interest in landscape research [11, 17, 69]. In line

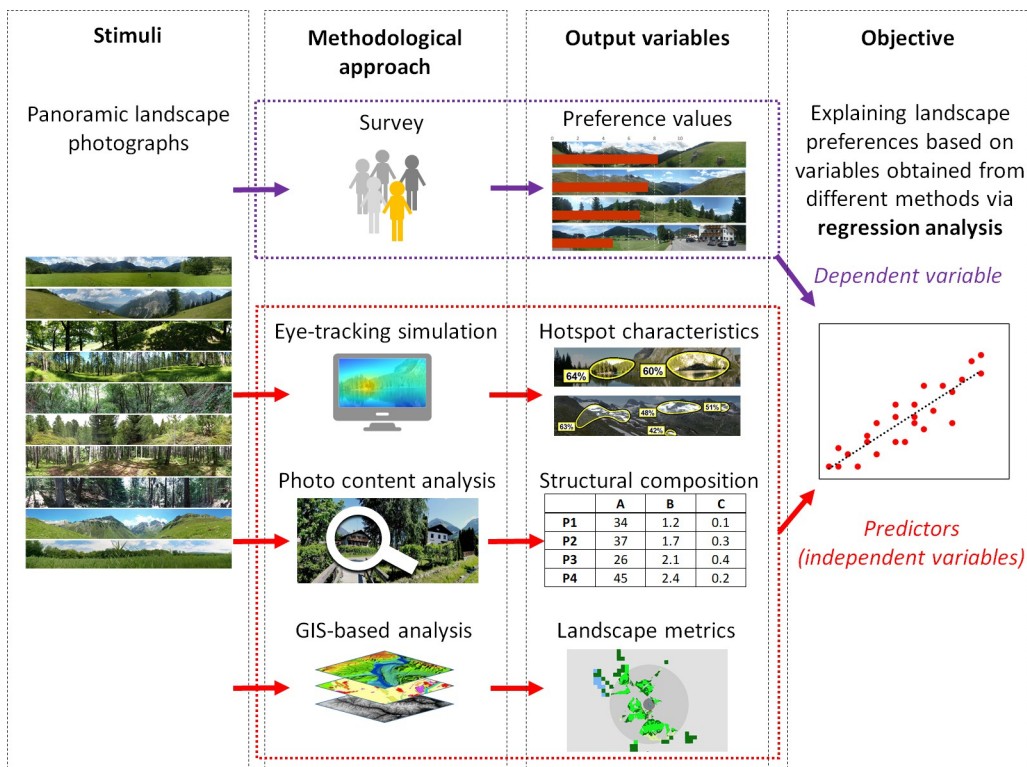

**Fig 1. Using eye-tracking simulation and spatial analysis to explain people's preferences of alpine landscapes, which were collected via surveys based on panoramic landscape photographs.** Photographs from Schirpke et al. [11], published under the Creative Commons CC BY 4.0 license.

with such typical consumer perceptual analyses, landscape analysis uses criteria for explaining preferences that focus primarily on structural features and visual aspects, knowledge on functional relationships or emotional relationships [26, 34, 70, 71]. Hence, a person's interests, emotions and socio-cultural background (i.e., all top-down attentional values) are used as predictor variables [34, 69, 72]. In contrast, this study aims to also introduce bottom-up factors into landscape analysis, which are common in consumer analyses. Bottom-up factors, or exogenous stimulus-driven attention, are related to involuntary allocation of attention elicited based on salient features such as luminance and size, coloration and contrasts (e.g., yellow-blue contrasts, light-shadow), brightness and orientation [68]. These exogenous stimuli are included in this study by using eye-tracking simulation to improve the prediction of people's landscape preferences.

Panoramic landscape photographs were used as stimuli, for which a set of predictor variables was collected (Fig 1). For this purpose, we used different approaches, including (1) an eye-tracking simulation to identify and analyze the most important zones of initial eye-tracking movements (as bottom-up variables), (2) a photo content analysis to estimate the structural composition of the photos such as percentage sky, cloud cover, Land Use/Land Cover (LULC) types, etc. (as top-down variables), and (3) a Geographic Information System (GIS)-based analysis to measure composition and configuration (i.e., landscape metrics) of the visible landscape (as top-down variables).

To spatially structure the photo content analysis as well as to prepare and analyze the data used in the GIS-based analysis, accounting for the influence of distance on distinguishability/ recognizability of landscape features and LULC types [3, 13, 73], we used four distance zones:

1. Surrounding zone (Zone 1; 0–60 m): Individual landscape features can be clearly identified by the observer [73]

2. Near zone (Zone 2; 0.06–1.5 km): Individual landscape features such as trees or buildings are clearly discernible.

3. Middle zone (Zone 3; 1.5–10 km): Individual landscape features are not discernible anymore but different LULC types such as forest, grassland, settlement areas can be clearly identified.

4. Far zone (Zone 4; 10–50 km). Few major LULC types are distinguishable. A good visibility outside population centers in is mostly limited to about 40–50 km [74].

The photographs covered a variety of major LULC types of the Central Alps such as settlement areas, arable land, forest, mountain grassland, moors, rivers and lakes, as well as rocky high-mountain landscapes, partly covered by glaciers (Fig 2).

## Preference surveys

We combined the results of three surveys that gathered people's preferences of Alpine landscapes using photo-based questionnaires [13, 17, 75] to cover a high variety of different

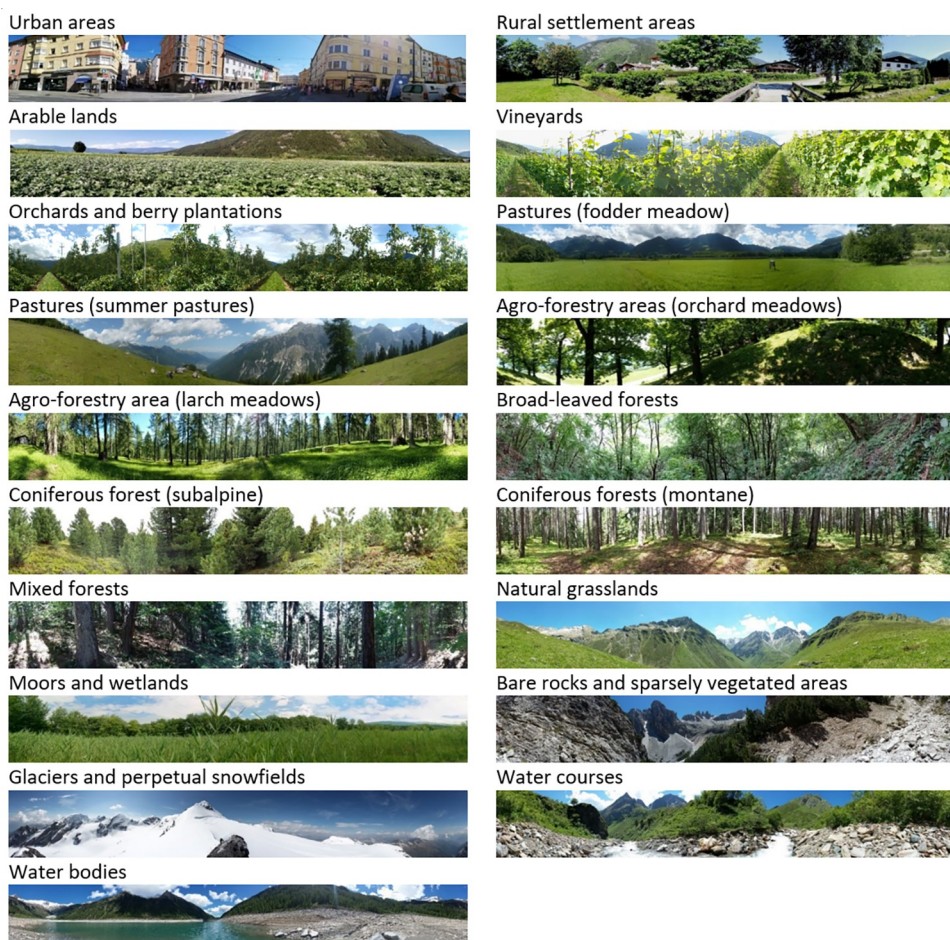

**Fig 2. Examples of photographs representing 19 different LULC types.** Photographs from Schirpke et al. [11], published under the Creative Commons CC BY 4.0 license.

landscapes. To assure comparability of the results [76, 77], all three questionnaires were designed and structured in a similar way in terms of picture format, rating type, sampling approach and study area, while partly focusing on different LULC types such as high-mountain landscapes or agricultural landscapes. Moreover, sixteen photographs of third survey [75] were taken from the two previous surveys [13, 17], i.e., from each survey 8 photographs, to enable the alignment of the preference scores of all three surveys to the result of Forer et al. [75]. All questionnaires included 360˚ panorama photographs that were taken at sunny days during summer at normal eye level (approx. 1.5–1.7 m). The photographs were randomly arranged in a paper-based questionnaire (A4 landscape orientation, max. 4 photographs on one page) and printed with high resolution. In two surveys [13, 75], people's preferences were assessed by asking the respondents to indicate their preferences on a 10-point Likert scale (1 = 'least preferred' to 10 = 'most preferred'). The other survey [17] used a 5-point Likert scale (1 = 'least preferred' to 5 = 'most preferred') and the results were rescaled into a 10-point Likert scale applying linear stretch method [77]. Moreover, questions on socio-demographic information such as gender, age, and nationality were collected. The questionnaires were available in German and/or Italian. All surveys were carried out in the Central Alps between 2011 and 2019, applying a stratified sampling approach to include both residents and visitors, accounting for demographics and origin. The first survey [13] comprised 858 respondents, the second survey [17] 384 respondents and the third survey [75] 967 respondents.

Since all previous analyses demonstrated that differences in the preference scores between different socio-demographic groups were small [11, 13, 17, 75], mean preference scores were calculated for all photographs after aligning the rating scale and the preference scores among the three surveys based on the linear stretch method [77]. From the entire pool of 212 photographs with at least 15 pictures per LULC type (see Table 2), we selected 78 photographs for this study, depicting different landscapes taken in different locations (S1 Fig). For each LULC type in the study area, we chose 4–5 photos to represent the diversity of spatial characteristics of the different LULC types. The selection followed a stratified sampling approach by stepwise integrating the following criteria: 1) different land-use intensity, 2) topography (with special attention to the slope), and 3) landscape context (see Zoderer et al. [17]). We included only photos, for which the horizon did not extent over the center of the picture, and which had a neutral sky (e.g., no threatening cloud atmosphere, no sunrise or sunset, no special sky colors) to reduce potential influence from the appearance of the sky on the preference scores [78, 79].

## Collecting predictor variables

**Eye-tracking simulation analysis.** We used 3M-VAS (https://vas.3m.com) to analyze initial eye-tracking movements on the photographs and thus to specifically bring factors affecting top-down attention into the analysis. The panoramic photographs were already standardized for the questionnaires and not altered for upload in 3M-VAS, i.e., all images had the same size (2300x360 pixel) and resolution (300 dpi). 3M-VAS does not provide much information about quality standards, but there is a limit in 'nominal' resolution, under which the system will warn that it is insufficient, which was not the case for our photographs. We scanned all photographs, using the category "Other" as the most general and unbiased modality [59]. The analysis report of 3M-VAS always included a heatmap, hotspots, a gaze sequence (most probable viewing order of the first four points) and a report of visual elements (i.e., edges, intensity, red-green color contrast, blue-yellow color contrast, faces) that have been used to calculate the heatmap (Fig 3). The output images of 3M-VAS had all a size of 1024x160 pixel and a resolution of 96 dpi. The software treats each photograph regardless of the others and results can be

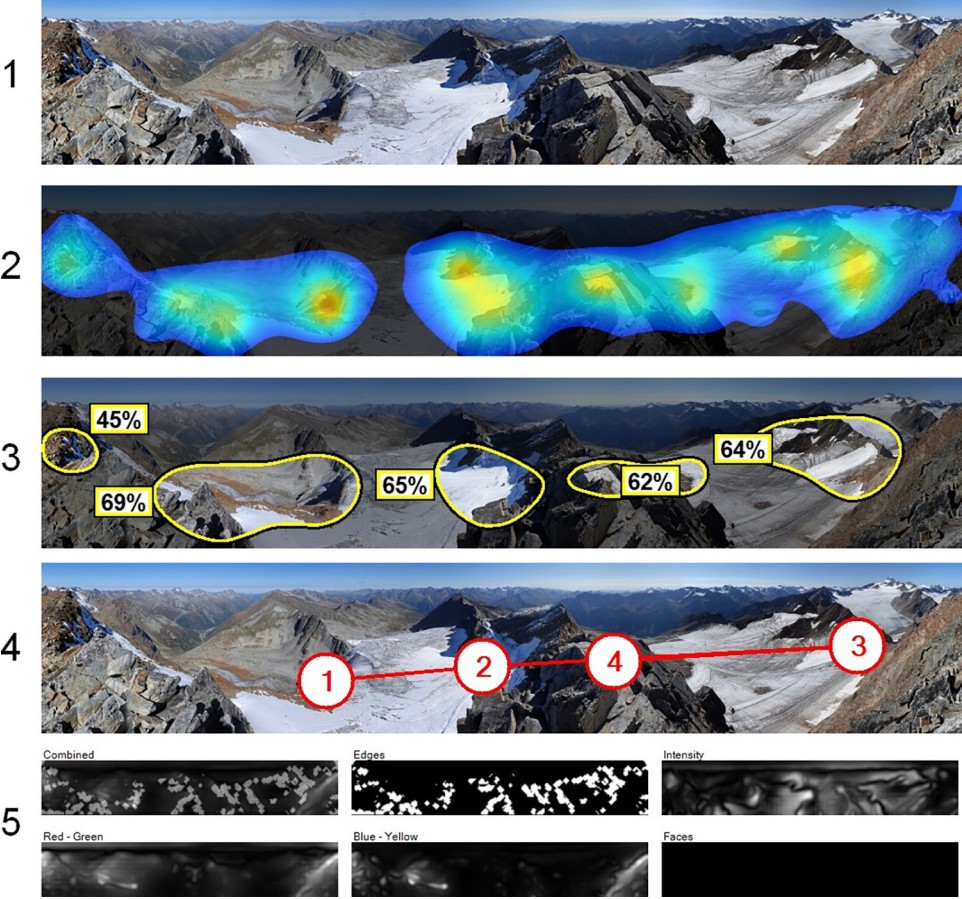

**Fig 3. Output of the 3M-VAS software.** (1) Original image, (2) Heatmap indicating the probability that areas are seen within the first 3–5 seconds, (3) Hotspots derived from the heatmap, specifying the probability that a person will look somewhere within the hotspot areas within the first 3–5 s), (4): Gaze sequence (most probable viewing order) of the 4 most-likely seen gaze locations, (5) Visual elements (edges, intensity, red-green color contrast, blue-yellow color contrast, faces), indicating how each of the elements contributes to the overall probability. Photograph by E Tasser.

considered as independent. Whatever limitations may be a common denominator within all photographs, so they should not influence comparative results.

We used the results of the hotspots analysis to derive different variables for each photograph (Table 1). First, we extracted primary hotspot characteristics, i.e., quantitative and spatial information generated by 3M-VAS such as the number of hotspots and probabilities of initially viewing the hotspots. Then, we also estimated secondary hotspot characteristics, i.e., variables in relation to the different distance zones as well as natural and artificial features within the hotspots, e.g., sky, cloud, snowfield/glacier, lighting conditions, specific natural or anthropogenic features. This evaluation was carried out for both the top-hotspots (hotspots with the highest probability of initial eye-tracking per photo) alone, as well as across all hotspots.

Furthermore, we analyzed the contribution of the visual elements (edges, intensity, redgreen color contrast, blue-yellow color contrast, faces; see Fig 3.5) for the hotspots to better understand the importance of these different elements. For this purpose, we estimated the contribution of the individual elements to the overall probability within the hotspot areas on a scale of from 0 to 100%, taking into account the share of the area and the combined value levels. For example, if edges took up 50% of the area within a hotspot, intensity had mostly

**Table 1. Variables extracted from the hotspots that were identified by 3M-VAS.**

| Group | Variable | Unit | Description |
|---|---|---|---|
| Primary hotspot characteristics | HP | (n) | Number of hotspots within the photo (see Fig 3.3) |
| | $HP_{area}$ | (%) | Total estimated area of the hotspots in the photo |
| | $HP_{mean}$ | (%) | Mean probability of initially looking at the hotspots, mean across all hotspots |
| | $HP_{mean\_w}$ | (%) | Area weighted mean probability of initial eye-tracking movement, mean of all hotspots |
| | $HP_{max}$ | (%) | Probability of initial eye-tracking movement of the hotspot with the highest probability (top-hotspot) |
| | $HP_{min}$ | (%) | Probability of initial eye-tracking movement of the hotspot with the lowest probability |
| | $HP_{max\_area}$ | (%) | Estimated area of the top-hotspot |
| Secondary hotspot characteristics | $HP_{Z1}$ | (n) | Number of hotspots within the surrounding zone (Z1; 0–60 m) |
| | $HP_{Z2}$ | (n) | Number of hotspots within the near zone (Z2; 0.06–1.5 km) |
| | $HP_{Z3}$ | (n) | Number of hotspots within the middle zone (Z3; 1.5–10 km) |
| | $HP_{Z4}$ | (n) | Number of hotspots within the far zone (Z4; 10–50 km) |
| | $HP_{1..n}$ | (%) | Estimated area of individual LULC types within the hotspots |
| | $Sky\_HP_{1..n}$ | (%) | Estimated area of sky within the hotspots |
| | $Cloud\_HP_{1..n}$ | (%) | Estimated area of clouds within the hotspots |
| | $Light\_HP_{1..n}$ | (%) | Estimated area of special lighting phenomena within the hotspots |
| | $Snow\_HP_{1..n}$ | (%) | Estimated area of snow/glacier within the hotspots |
| | $E_{nat}\_HP_{1..n}$ | (%) | Estimated area of individual natural elements in the foreground within the hotspots (e.g., stones, flowers, plants) |
| | $E_{art}\_HP_{1..n}$ | (%) | Estimated area of individual artificial elements in the foreground within the hotspots (e.g., streets, street signs, cars, fences) |

medium to high values (grey to light grey areas), low red-green contrast values (dark grey patterns) and missing blue-yellow contrasts as well as no values for faces (black), such a hotspot received a value of 50 for edges, 40 for intensity, 5 for red-green color contrast and 0 for blue-yellow color contrast and faces (see S2 Fig, hotspot no. 3). For each photograph, these values were averaged over all existing hotspots.

All estimations were performed by one of the authors to avoid uncertainty in the area estimates by different persons. This person first gridded some photographs for own calibration and then derived the proportions of certain features by the number of grids. This approach was discussed and determined in advance by the author team.

**Photo content analysis.** To quantitatively describe the content of the photos, we used eight indicators (Table 2), which were mainly related to the type of LULC and composition of

**Table 2. Variables extracted from visual photo content analyses.**

| Variable | Unit | Description |
|---|---|---|
| Z | (n) | Number of distance zones in the photo (surrounding zone (0–60 m), near zone (0.06–1.5 km), middle zone (1.5–10 km), far zone (10–50 km)) |
| $LULC\_P_{1-n}$ | (%) | Estimated area of visually well distinguishable LULC types (water bodies, water courses, moors and wetlands, coniferous forest (montane and subalpine), broad-leaved forest, mixed forest, glaciers and snowfields, pastures (summer pastures), pastures (fodder meadows), orchards and berry plantations, vineyards, arable lands, urban areas, rural settlement areas) within the photo |
| $E_{nat}\_P_{1..n}$ | (%) | Estimated area of clearly recognizable natural elements within photo (e.g., stones, flowers, single plants) |
| $E_{art}\_P_{1..n}$ | (%) | Estimated area of clearly recognizable artificial elements within the photo (e.g. street, street signs, cars, fences) |
| Sky | (%) | Estimated area of sky within the photo |
| Clouds | (%) | Estimated area of clouds within the sky |
| Light | (%) | Estimated area of special lighting conditions within the photo |
| Open soil | (%) | Estimated area of open soil cover |

the photos and which may positively or negatively influence people's preferences [73, 80]. For example, natural ecosystems and landscape features are generally preferred over intensively used ecosystems and artificial elements [16–18]. Therefore, we included the proportion of LULC types in the photo and estimated the proportion of clearly recognizable human structures such as roads, paths, fences and signs. Furthermore, the composition of a photo in terms of position of horizon and appearance of the sky can have positive or negative effects on preferences [78, 79]. Hence, we estimated the proportion of the sky in the photos and the proportion of cloud cover in the sky. In addition, the proportion of areas with special lighting conditions such as extreme changes in shade and sun or reflections of partial landscapes in a lake were recorded. This approach was also discussed in advance by the author team, and the estimates were subsequently performed by only one author.

**GIS-based analysis.** Landscape characteristics of visible landscape in form of landscape metrics were obtained from Schirpke et al. [11]. Landscape metrics provide a quantitative framework to describe spatial landscape pattern, i.e., composition and configuration, which has also been related to landscape preferences [24–26, 31, 81, 82]. Landscape metrics were calculated based on a mosaic of LULC maps (spatial resolution of about 27.4 x 27.4 m) that included only the visible area, as some areas on the map may be hidden due to the topography of mountain landscapes. The visible area up to 50 km [74] seen from the photo location was identified through viewshed analysis that determines for each cell of the DSM whether it is within the observer's line-of-sight or not [83]. Viewhsed analysis was based on digital surface models (DSM), which had different spatial resolution (S1 Table) for the four distance zones (see above). We overlaid the visible area with LULC maps with different spatial and thematic resolution to account for the influence of increasing distance from the photo location on distinguishability/recognizability of landscape features and different LULC types in terms of perceived size and color [3]. This means that, for example, a berry plantation would be a specific habitat type in zones 1 (at the observer point) and 2 (near zone), while it would not be distinguishable in zones 3 (middle zone) and 4 (far zone), i.e., it is merged with other land cover types, presenting a separately recognizable area without knowing what exactly it is. Therefore, LULC maps were selected and prepared according to the four distance zones (S1 Table). After overlaying the visible area with the LULC maps, we created mosaic of the visible LULC, which was used to calculate landscape metrics. Non-visible areas were classified as background to exclude their influence on area-based metrics. Landscape metrics included median of patch area (AREA_MD), standard deviation in related circumscribing circle distribution (CIRCLE_SD), Largest patch index (LPI), modified Simpson's diversity index (MSIDI), number of patches (NP), patch richness (PR), i.e., the number of different the LULC types present, patch density (PD), median of contiguity index (CONTIG_MD), standard deviation shape index (SHAPE_CV), median of gyration radius (GYRATE_MD) and total area of zones 1 and 2 (TA_1, TA_2). For full details, see Schirpke et al. [11].

## Statistical analysis

To explain people's preferences by independent variables derived through the three different approaches, we used backward stepwise linear regression models (Fig 1). In particular, we aimed to identify the influence of the variables obtained from the eye-tracking simulation and related hotspot analysis. We therefore set up four different models, progressively increasing the number of predicting variables:

1. In the first regression model, only variables that described primary hotspot characteristics resulting from the eye-tracking simulation were used as predictors (see, Table 1).

2. In the second regression model, the secondary hotspot characteristics resulting from the interpretation regarding distance zones and natural and artificial features were additionally included to the primary hotspot characteristics (see, Table 1).

3. In the third regression analysis, only the variables extracted by photo content analysis (Table 2) and landscape metrics from GIS-based landscape analysis (see previous section) were used as predictors.

4. In the fourth regression analysis, all variables (photo content analysis, landscape metrics, all variables describing hotspot characteristics) were used to explain preference value dispersion (see Schirpke et al. [11, 13]).

In total, 8 primary and 11 secondary hotspot characteristics (Table 1), 18 photo content characteristics (Table 2) as well as 12 landscape metrics were used. Missing values for predictors were replaced with mean values. To avoid overfitting and facilitate model interpretability, we eliminated unnecessary and collinear predicting variables by a two-step process. First, we screened for multicollinearity in the predictor set by calculating tolerance and variance inflation factor (VIF) values for each variable and only included those with a tolerance $>0.1$ and a variance inflation factor (VIF) $<10$. Second, a backward stepwise linear regression routine was applied with the remaining variables to further reduce the number of variables [84], starting with all variables and iteratively removing the weaker predicting variables until no further improvement was possible. Finally, we interpreted the adjusted $R^2$, because it is less affected by overfitting than $R^2$, and thus, it is easier to assess how much variability the model really 'explains' [85]. The difference between $R^2$ and the adjusted $R^2$ was used as a final estimate of the degree of overfitting.

Validity, quality and significance of the coefficients were determined by analysis of variance (ANOVA) or t-tests. In addition, we checked the reliability of the regression using the Shapiro-Wilk test (residual normality), the Durbin-Watson test (autocorrelation of the variables), the Breusch-Pagan test (homoscedasticity) and the Tolerance/VIF statistics (multicollinearity effect). All statistical analyses were performed in SPSS Statistics (IBM SPSS 27).

## Results

### Landscape preferences

The mean preference values differed significantly between different LULC types (Fig 4, S2 Table). Lakes and high alpine landscapes with glaciers and snowfields had highest preference values, followed by agro-forestry areas such as larch meadows and alpine summer pastures. Alpine grassland, bare rock landscapes, water courses, broad-leaf forest and semi-open coniferous forests in the subalpine belt obtained medium preferences. The lowest preference values received urban areas, permanent crops as orchards and vineyards and dense coniferous forests in the montane belt.

### Predictor variables

**Characteristics of hotspots.**  3M-VAS identified on average $3.3 \pm 1.1$ (s.d.) hotspots (HP) per photograph. The mean estimated proportion of hotspots ($HP_{area}$) was $17.4 \pm 6.3\%$ of the photo area. The mean probability of initially looking at the hotspots ($HP_{mean}$) was $61.9 \pm 6.7\%$ (area weighted mean: $65.6 \pm 7.1\%$), the probability of initial eye-tracking movement of the top-hotspot ($HP_{max}$) $74.1 \pm 10.1\%$ (mean $HP_{max\_area}$: $7.5 \pm 3.0\%$ of the photo area) and the probability of the lowest hotspot ($HP_{min}$) was $50.6 \pm 9.4\%$. With regards to the secondary hotspot characteristics, the descriptive analysis of the hotspots indicated elements that were most likely to be seen within the first 3–5 s (Fig 5). More than 80% of the top-hotspots contained single trees and tree trunks, more than 70% buildings and horizon transitions, if these contents

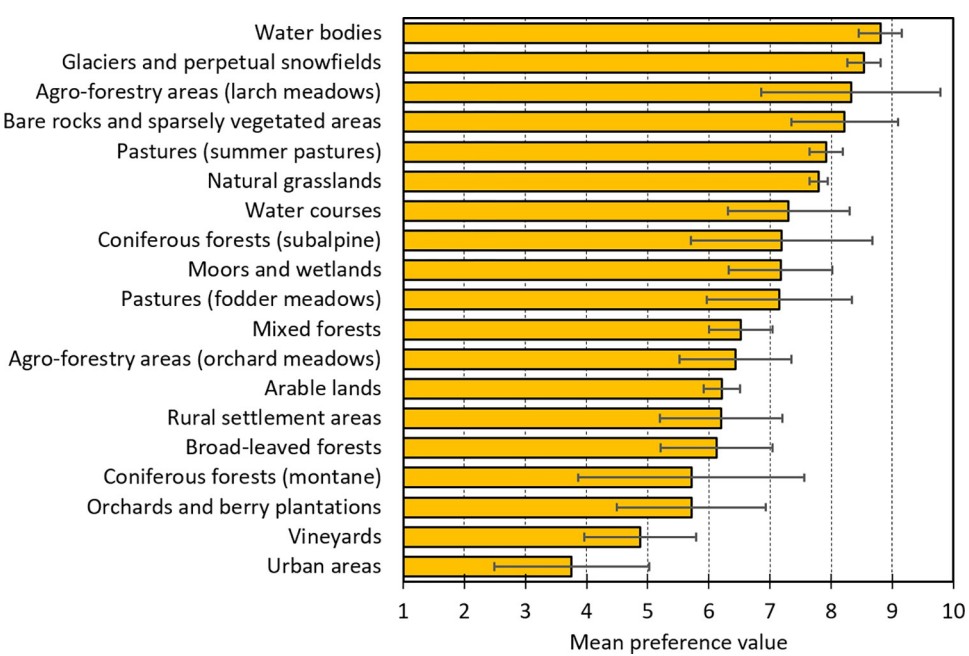

**Fig 4. Mean preference values ($\bar{x} \pm s.d.$, n = 4–5) of different LULC types, ranging from 1 = 'least preferred' to 10 = 'most preferred' (Likert scale).**

appeared in the photo. These photo contents were viewed first, i.e., a correspondingly high proportion already occurred in the top hotspots. With a probability of more than 50%, glaciers or snowfields, rocks, as well as agriculturally used grassland patches were also included in the top-hotspots. In addition, with longer viewing, i.e., considering all hotspots in the photos, the probability of alpine grasslands and forests strongly increased. Strong increases in viewing probability also occurred for mountain peaks, rocks und scree slopes as well as lakes and rivers. Individual artificial elements (e.g., fences, street signs, cars), individual natural elements (e.g. plant leaves, single stones), clouds and special light effects (e.g., change of sun and shadow, reflections in the lake) were much less common.

In general, the highest contribution of the visual elements to the hotspots had edges, followed by intensity, red-green color contrast and blue-yellow color contrast, while faces had no importance for (semi-)natural landscapes (Fig 6). Among the different LULC types, the contributions of the individual elements partly varied. For orchards and berry plantations, orchard meadows, urban areas, montane coniferous forest as well as glaciers and perpetual snowfields, edges were most important, but color contrast were below average. On the contrary, edges were less important particularly for summer pasture, natural grasslands as well as moors and wetlands, while the contribution of color contrasts was above average for these LULC types.

## Photo content and landscape characteristics

The photos differed in terms of content, structure and elements depicted (see S3–S5 Tables). Photos with high-elevated LULC types (e.g., glaciers and rocks, alpine grasslands) were characterized by an above-average number of visible distance zones (Z) and an above-average proportion of sky area (sky) due to the open view. Accordingly, some landscape metrics (AREA_AM, LPI, PR and GYRATE_MD) were above average compared to photos with less distance zone due to the presence of higher vegetation, e.g., photos taken within different forest types mostly including only one zone. A characteristic of many forest types (especially

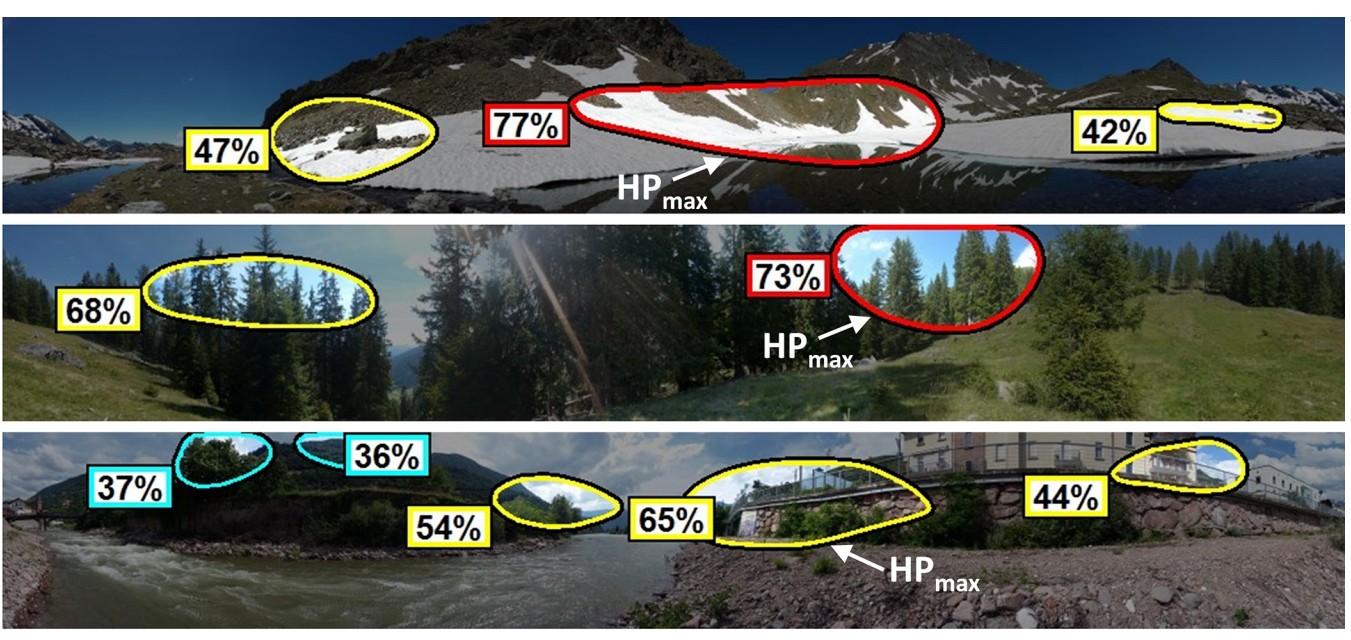

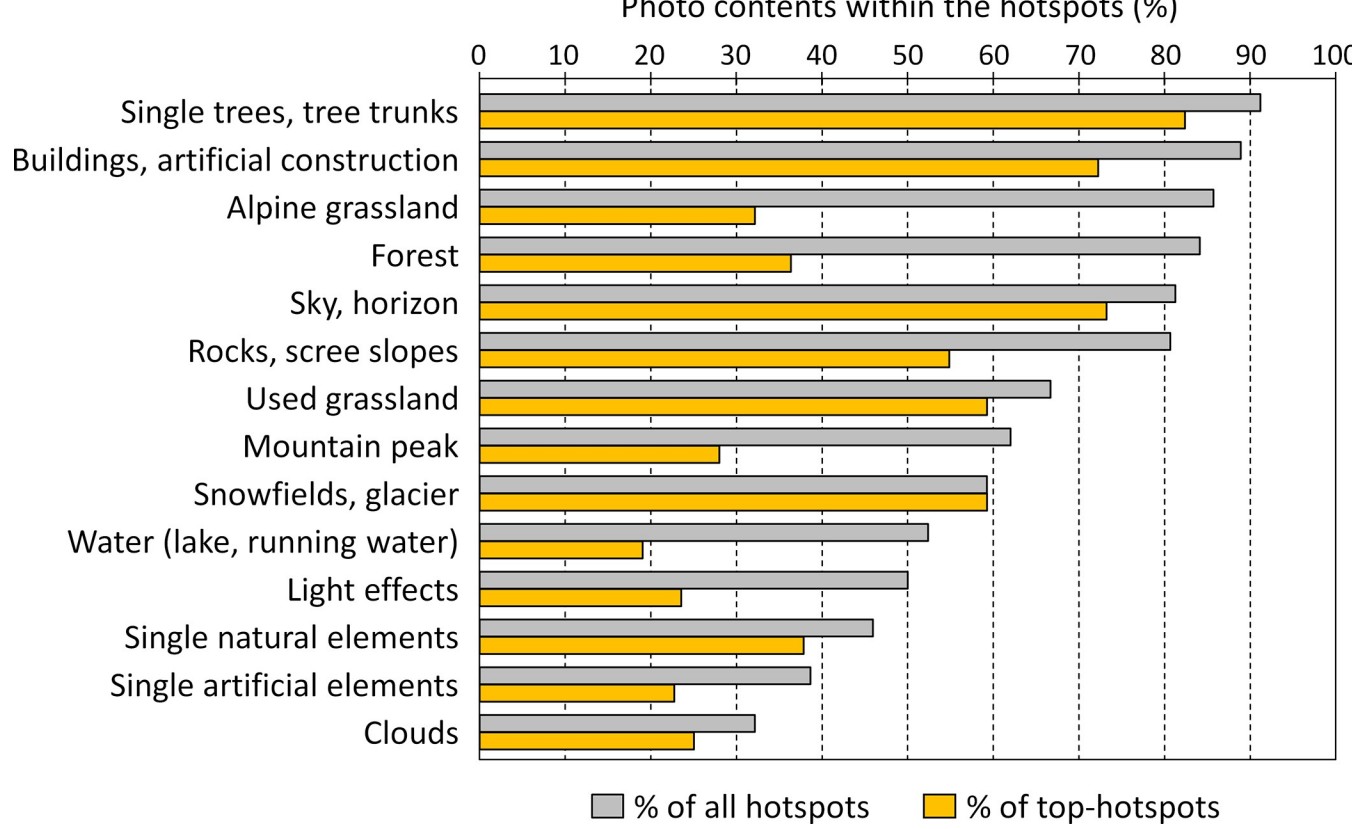

**Fig 5. Examples of identified eye-tracking hotspots by 3M-VAS and the frequency of photos with at least one specific content within the top-hotspots (HP$_{max}$) and all hotspots.** Frequencies were calculated considering only those photos containing the specific contents, e.g., number of hotspots with water in relation to all photos including hotspots with water. Photographs from Schirpke et al. [11], published under the Creative Commons CC BY 4.0 license.

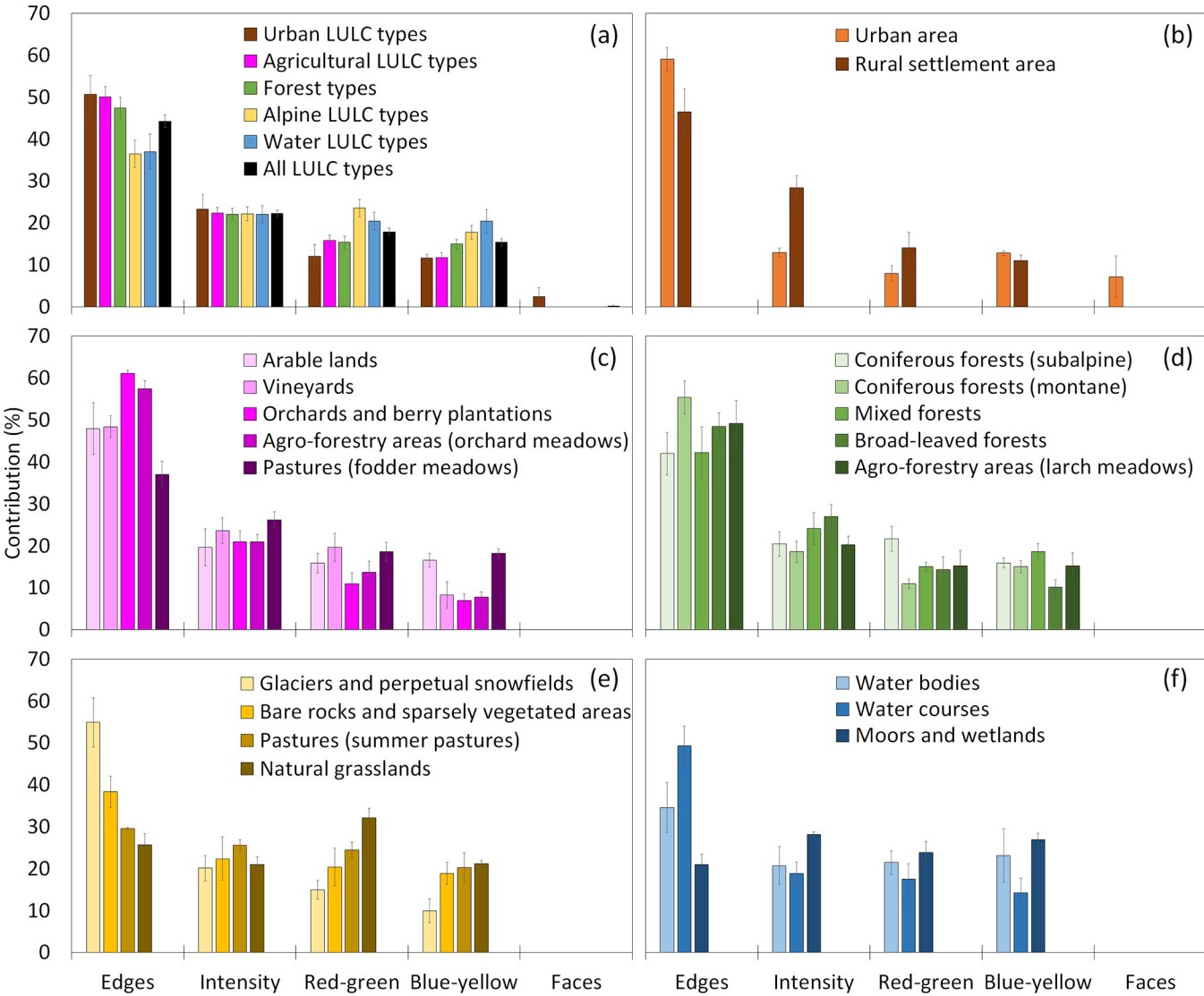

**Fig 6.** Contribution of visual elements (edges, intensity, red-green color contrast, blue-yellow color contrast, faces) to the probability that hotspot areas are seen within the first 3–5 seconds across all LULC types (a), urban LULC types (b), agricultural LULC types (c), forest types (d), alpine LULC types (e) and water LULC types (f).

mixed and broad-leafed forests) was the low vegetation cover on the forest underground, caused by the lack of light in these dense forest canopies. In contrast, rural settlement areas and agricultural landscapes close to settlements were characterized by a high proportion of artificial elements and a high diversity of different LULC types. The photos including rivers also included many different LULC types.

## Prediction of landscape preferences

To analyze the influence of hotspots on landscape preferences, three regression analyses with different predictors were performed. The first analysis using only primary hotspot predictors indicated that neither the area of the hotspots nor the resulting probability values were suitable to explain landscape preferences (ANOVA p = 0.072; $R^2$ = 0.042; adjust. $R^2$ = 0.029). Adding the secondary hotspot indicators significantly increased the explanatory value (p = 0.000; $R^2$ =

**Table 3. Result of the backward stepwise linear regression, including only eye-tracking predictors with tolerance >0.1 and variance inflation factor (VIF) <10 during collinearity diagnostics.** Only variables with p<0.1 are shown.

| Variables | Non standardized coefficient | | Standardized coefficient Beta | T | Sig. | Collinearity statistics | |
|---|---|---|---|---|---|---|---|
| | Regression coefficient $B$ | SD | | | | Tolerance | VIF |
| (Constant) | 6.810 | .373 | | 18.259 | .000 | | |
| HP (n) | -.288 | .119 | -.215 | -2.415 | .018 | .788 | 1.269 |
| $Snow\_HP_{1..n}$ (%) | .035 | .019 | .150 | 1.824 | .072 | .926 | 1.080 |
| $E_{art}\_HP_{1..n}$ (%) | -.048 | .009 | -.429 | -5.286 | .000 | .951 | 1.051 |
| $HP_{Z2}$ (n) | .729 | .125 | .495 | 5.852 | .000 | .873 | 1.146 |
| $HP_{Z3}$ (n) | .646 | .144 | .400 | 4.497 | .000 | .790 | 1.266 |
| $HP_{Z4}$ (n) | .458 | .244 | .156 | 1.878 | .065 | .902 | 1.108 |

0.556; adjust. $R^2$ = 0.519; Table 3). There was weak evidence that the proportions of snowfield/glacier areas ($Snow\_HP_{1..n}$) and the number of hotspots within the far zone of the photo ($HP_{Z4}$) had positive effects on the preference value (p = 0.05–0.1), while a moderate evidence (p = 0.018) was found for the number of hotspots within the photo (HP) with a negative effect. Very strong evidence was found that the presence of artificial elements ($E_{art}\_HP_{1..n}$) is negatively associated and the number of hotspots in the near zone ($HP_{Z2}$) and in the middle zone ($HP_{Z3}$) of the photo were positively associated with the preference value (p<0.001).

The third analysis, including all variables from the photo content analysis and the GIS-based analysis without the hotspot indicators, led to a model explanation of 59% of the variability in preferences (p = 0.000; $R^2$ = 0.612, adjust. $R^2$ = 0.585), suggesting that these photo attributes had a lower explanatory value than the hotspot indicators. It shows only a moderate evidence (p = 0.034) for median of patch area (AREA_MD). The fourth analysis, including all hotspot indicators as well as all variables from the photo content analysis and the GIS-based analysis, led to the improvement of the model explanation from 59% of the variability in preferences to 83% (p = 0.000; $R^2$ = 0.873, adjust. $R^2$ = 0.834, see Table 4). There was very weak evidence that the estimated area of the top-hotspot ($HP_{max\_area}$) decreased landscape preferences. With moderate evidence, the number of hotspots within the near ($HP_{Z2}$) and middle zone of the photo ($HP_{Z3}$) had positive effects, whereas the number of hotspots within the far zone of the photo ($HP_{Z4}$) negatively affected landscape preferences. The data revealed strong evidence that the probability of initial eye-tracking movement of the hotspot with the lowest probability ($HP_{min}$), the number of zones (Z), the largest patch index (LPI) and the total area of near zones (TA_2) were positively associated. Furthermore, the increase in the proportion of broad-leaved forests led to an increase of preferences, whereas an increase of the proportion of moors and wetlands and of urban areas & rural settlement resulted in a decline of preferences. Very strong evidence (p<0.001) was found that the proportion of sky and water bodies & water courses, the proportion of mixed forests and larch meadows as well as the proportion of open soil were positively associated with the preference value, while the proportion of arable land was negatively associated with the preference value. Differently to the results of the second regression analysis (Table 3), there was no evidence for snowfield/glacier area ($Snow\_HP_{1..n}$).

## Discussion

**Characteristics of the hotspot analysis derived from eye-tracking simulation.** The identified hotspots mostly contained single trees and tree trunks, buildings and horizon transitions, which corroborates the results of other studies [86, 87]. Other elements such as glaciers, snowfields and rocks were also often included in the hotspots, representing all elements with a high

**Table 4. Result of the backward stepwise linear regression, including landscape metrics, photo content indictors and hotspot predictors with tolerance >0.1 and variance inflation factor (VIF) <10 during collinearity diagnostics.** Only variables with p<0.1 are shown.

| Variables | Non standardized coefficient | | Standardized coefficient Beta | T | Sig. | Collinearity statistics | |
|---|---|---|---|---|---|---|---|
| | Regression coefficient B | SD | | | | Tolerance | VIF |
| (Konstante) | 3.767 | .521 | | 7.234 | .000 | | |
| $HP_{max\_area}$ (%) | -.048 | .026 | -.104 | -1.877 | .066 | .698 | 1.432 |
| $HP_{min}$ (%) | .023 | .008 | .153 | 2.841 | .006 | .738 | 1.354 |
| $E_{art}\_HP_{1..n}$ (%) | -.046 | .006 | -.419 | -7.691 | .000 | .726 | 1.378 |
| $HP_{Z2}$ (n) | .221 | .097 | .150 | 2.277 | .026 | .497 | 2.012 |
| $HP_{Z3}$ (n) | .198 | .098 | .123 | 2.015 | .048 | .582 | 1.717 |
| $HP_{Z4}$ (n) | -.406 | .173 | -.138 | -2.345 | .022 | .619 | 1.614 |
| Z (n) | .332 | .112 | .246 | 2.955 | .004 | .311 | 3.212 |
| Sky (%) | .031 | .007 | .267 | 4.719 | .000 | .674 | 1.484 |
| LPI | .025 | .008 | .215 | 3.037 | .004 | .432 | 2.317 |
| TA_2 | .000 | .000 | .203 | 2.942 | .005 | .452 | 2.211 |
| Water bodies & water courses (%) | .040 | .009 | .218 | 4.195 | .000 | .795 | 1.257 |
| Moors and wetlands (%) | -.030 | .008 | -.192 | -3.524 | .001 | .724 | 1.381 |
| Broad-leaved forests (%) | .013 | .004 | .191 | 3.093 | .003 | .563 | 1.776 |
| Mixed forests (%) | .018 | .004 | .255 | 4.213 | .000 | .589 | 1.697 |
| Agro-forestry areas (larch meadows) (%) | .025 | .004 | .338 | 5.652 | .000 | .603 | 1.658 |
| Arable lands (%) | -.028 | .005 | -.305 | -5.673 | .000 | .743 | 1.346 |
| Urban areas & rural settlement areas (%) | -.017 | .006 | -.149 | -2.871 | .006 | .803 | 1.246 |
| Open soil (%) | .031 | .007 | .267 | 4.719 | .000 | .674 | 1.484 |

degree of naturalness, which are generally preferred over artificial elements [26, 33, 47]. In contrast to other studies [47, 86], single artificial elements (e.g., fences, street signs, cars) or individual natural elements (e.g., plant leaves, single stones) were hardly tracked in our study, which may be because certain elements such as benches are more in the focus of people's interests and therefore more recognized in real eye-tracking experiments [87]. It has to be remembered that 3M-VAS only tracks initial fixations, which reflect information gathered pre-attentively, so such cognitive parameters are not within its reach; assuming the claimed number of 92% accuracy is correct, our result should not differ to those that would be recorded with real eye-tracking of early glances. This contrasts with longer recording periods of the studies mentioned above (120 s acquisition period for image exploration [47] and monitoring during the complete 3 m 50 s of the video presentation [38]). Further information from real eye-tracking experiments, such as fixation count and duration, is currently not implement in 3M-VAS, but such variables may provide important insights, for example, into people's difficulty in extracting information or interpreting a landscape [30]. Nevertheless, our findings on the importance of the underlying visual elements for the hotspots indicate that edges have the highest contribution to the hotspots in (semi-)natural landscapes, while color contrasts are less important, which is, for example, in contrast to consumer science related to packaging, label design and advertisements [68]. Furthermore, clouds, special light and color effects (e.g., luminance, change of sun and shadow, reflections in the lake, edge contrast) were less important in our study. Although such low-level image properties can influence people's perceptions [43], people usually focus on the foreground and central zones instead of areas with the highest image salience [28, 88].

Landscape preferences can depend on the quality and distribution of the "interesting" objects [33, 55, 86], but frequently viewed elements indirectly contribute to the perception

process, as the observer forms an opinion about these hotspots. For example, Cottet et al. (2018) found that observers mentioned the same objects that were in the hotspots when describing their motivations for judging a landscape. However, the selection of hotspots is significantly influenced by the type of search. During an active search, for example, visual saliency plays only a minor role, as the search for and the viewing of elements are mainly controlled by cognitive factors [89]. In the case of a passive search, a different process is assumed, although there is not yet sufficient evidence for this.

## Potential of eye-tracking simulation in landscape preference research

Adding the hotspot indicators increased the model explanatory power by 24%, which confirmed the results of another study by Wang et al. [90]. This increase may be explained by the relationships between fixation count and duration with landscape complexity, i.e., higher fixation counts but lower fixation duration with a higher level of landscape complexity [33, 47, 91]. In line with Li et al. [51], our results also indicate that the frequency of artificial elements led to a statistically significant reduction in preferences. Furthermore, eye-tracking on the level of cultivation (i.e., degree of abandonment/stage of succession, presence of weeds, type/frequency of management) and on the status and condition of human-made structures (status and maintenance of structures such as fences and farm buildings) can provide important information to evaluate stewardship [43].

Our findings indicate that the analysis of eye-tracking hotspots can support the identification of important elements and areas of a landscape, but they do not reveal people's preferences, which are largely determined by general landscape characteristics [25, 30, 33, 55]. This is not surprising, given the pre-attentive nature of the processing that guides the deployment of initial attention; this deployment is based on the presence of certain features, and can have a survival value, regardless of whether the stimulus is eventually judged as positive or not, on a conscious level. A more detailed discussion about the correlation of visual organization to 3M-VAS results for artificial structures has been done by Lavdas et al. [59]. With regard to landscape preferences, specific characteristics such as naturalness of LULC types and landscape complexity [33, 55] are more important, i.e., lakes, glaciers and semi-natural agro-forestry areas such as larch meadows and alpine summer pastures higher preference values than urban areas, permanent crops and dense coniferous forests. Furthermore, the degree of openness, vegetation composition, viewing depth, visibility and heterogeneity significantly influence landscape preferences [25, 30, 92]. Eye-tracking studies indicate less fixation on homogeneous landscapes due to their rather unvaried character and are perceived as more restorative [55], whereas heterogeneous landscapes are more 'entertaining', which explains its stronger visual exploration [30]. Accordingly, landscape complexity (colors, textures, shapes, physical dimensions of elements, topography, and structures) plays an important role for describing visual character [38], but studies generally do not agree on its effect on people's preferences. For example, some studies indicate that landscapes with a medium level of complexity are preferred over monotonous or highly complex landscapes [33, 34]. In contrast, Kaplan et al. [93] postulated a linear relationship between landscape complexity and preference. The type of organization that this complexity follows seems to be a key factor, as there is accumulating data supporting the notion of preferential perception of fractally based patterns, i.e., patterns based on an ordered geometrical structure with an hierarchy of scales [94–96].

## Future research directions

As discussed above, the assessment of the characteristics of different LULC types with eye-tracking simulation software has several constraints, in particular, with regard to explaining

landscape preferences. Nevertheless, deeper insights into relationships between the landscape and human preferences may be obtained by pointing at specific landscape characteristics, for which differences can be revealed based on fixation count rather than duration. For example, different levels of landscape complexity have been compared in urban green spaces [33], but this may be applied also in more natural environments to identify the importance of color contrasts, shape variations, spatial arrangement of landscape elements and landscape structure. Here, the use of purposively altered photographs or digitally designed landscapes can be useful in real-world decision-making situations [97, 98]. An integration of eye-tracking simulation, especially in the phase of feasibility studies, may support the evaluation of negative impacts on landscape perceptions in addition to studies on economic, social and ecological effects. Impacts may originate, for example, from new infrastructure such as wind turbines and photo-voltaic panels to increase energy production from renewable resources [99]. It could be tested with eye-tracking simulation software, which colors or shapes of such infrastructure will be less visible (i.e., by reducing edges and color contrasts). Another example of application may be the evaluation of measures to improve the attractiveness of agricultural landscapes by adding flowering stripes that increase color contrasts [4, 100]. In particular, in the planning and development of commercial and residential areas, different stakeholder groups as well as experts from various disciplines such as ecologists and psychologists should be involved in the evaluation of different scenarios, as interventions have effects far beyond sectoral boundaries. Hence, the evaluation of different scenarios by eye-tracking simulation may provide important information for decision-making and landscape planning. However, an operational use should be facilitated by providing numerical outputs of the 3M-VAS software (i.e., our proposed primary hotspot characteristics and mean values for underlying visual elements) or an automatized post-processing.

Finally, differences between groups with differing socio-cultural characteristics such as gender, age, social and social and cultural background have been widely discussed in research on landscape preferences [4, 17, 24, 25, 34], but only few eye-tracking studies have addressed such differences. For example, studies found gender-related differences, as women seem to follow more often a systematic strategy and look at the landscape more intensively and with longer fixation times than men [37]. Accordingly, the increase in pupil diameter indicates a more intensive processing of memories of women, while men tend to get a quick overview of the landscape [101]. Another study related eye movements to human behavior, indicating that the level of nature relatedness could explain whether individuals looked more likely at trees than buildings [102]. The extent to which socio-cultural differences affect people's preferences and perceptions should be a focus of future eye-tracking studies, as it provides an objective method to identify differences in people's perceptions.

## Supporting information

**S1 Fig. Location of the 78 panoramic photographs in the Central European Alps.** Data sources: EEA (2016; 2019) and OpenStreetMap (https://www.openstreetmap.org).
(TIF)

**S2 Fig. Contribution of visual elements to the overall probability that areas are seen within the first 3–5 seconds.** The yellow circles indicate the hotspots identified in 3M-VAS, which were used for estimating the importance of each visual element within each hotspot on a scale of from 0 to 100%. For example, for hotspot no. 3, edges take up 50% of the area, intensity has mostly medium to high values (grey to light grey areas), red-green contrast values are low (dark grey patterns) and blue-yellow contrasts as well as no values for faces (black) are missing. Estimated contributions are 50% for edges, 40% for intensity, 5% for red-green color contrast

and 0% for blue-yellow color contrast and faces.
(TIF)

**S1 Table. Input data for GIS-based landscape analysis according to Schirpke et al. (2021).**
(PDF)

**S2 Table. Mean preference values of 19 LULC types derived from three surveys as indicated by Schirpke et al. (2021).**
(PDF)

**S3 Table. Visual photo content indicator values (mean; min-max) for photos mainly covered by different LULC types.**
(PDF)

**S4 Table. Estimated area of visually well distinguishable LULC types (mean; min-max) within the photos.** LULC 1: water bodies & courses (%); LULC 2: glaciers and snowfields (%); LULC 3: Rocks, screen slopes; LULC 4: moors and wetlands (%); LULC 5: coniferous forests (subalpine) (%); LULC 6: coniferous forests (montane) (%); LULC 7: mixed forests (%); LULC 8: broad-leaved forests (%); LULC 9: grasslands (alpine grasslands, summer pastures) (%); LULC 10: pastures (fodder meadows) (%); LULC 11: orchards and berry plantations (%); LULC 12: vineyards (%); LULC 13: arable lands (%); LULC 14: rural settlement areas (%); LULC 15: urban areas (%).
(PDF)

**S5 Table. Landscape metrics (mean; min-max) for photos mainly covered by different LULC types.**
(PDF)

## Acknowledgments

The authors are grateful to Kelly Canavan, Global Marketing Development Manager for VAS at 3M Company for allowing the use of 3M-VAS software.

## Author Contributions

**Conceptualization:** Uta Schirpke, Erich Tasser, Alexandros A. Lavdas.

**Data curation:** Erich Tasser, Alexandros A. Lavdas.

**Formal analysis:** Uta Schirpke, Erich Tasser, Alexandros A. Lavdas.

**Methodology:** Uta Schirpke.

**Visualization:** Uta Schirpke, Erich Tasser.

**Writing – original draft:** Uta Schirpke, Erich Tasser.

**Writing – review & editing:** Uta Schirpke, Erich Tasser, Alexandros A. Lavdas.

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
