## [Decision Letter · Decision Letter 0]

4 Jul 2022

PONE-D-22-12355Potential of eye-tracking simulation software for analysing landscape preferencesPLOS ONE

Dear Dr. Schirpke,

Thank you for submitting your manuscript to PLOS ONE. After careful consideration, we feel that it has merit but does not fully meet PLOS ONE’s publication criteria as it currently stands. Therefore, we invite you to submit a revised version of the manuscript that addresses the points raised during the review process.

We look forward to receiving your revised manuscript.

Kind regards,

Ji-Zhong Wan

Academic Editor

PLOS ONE

Journal Requirements:

2. We note that Figure S1 in your submission contain [map/satellite] images which may be copyrighted. All PLOS content is published under the Creative Commons Attribution License (CC BY 4.0), which means that the manuscript, images, and Supporting Information files will be freely available online, and any third party is permitted to access, download, copy, distribute, and use these materials in any way, even commercially, with proper attribution. For these reasons, we cannot publish previously copyrighted maps or satellite images created using proprietary data, such as Google software (Google Maps, Street View, and Earth). For more information, see our copyright guidelines: http://journals.plos.org/plosone/s/licenses-and-copyright.

   a. You may seek permission from the original copyright holder of Figure S1 to publish the content specifically under the CC BY 4.0 license. 

Natural Earth (public domain): http://www.naturalearthdata.com

3. We note that Figures 1, 2, 3 and 5 in your submission contain copyrighted images. All PLOS content is published under the Creative Commons Attribution License (CC BY 4.0), which means that the manuscript, images, and Supporting Information files will be freely available online, and any third party is permitted to access, download, copy, distribute, and use these materials in any way, even commercially, with proper attribution. For more information, see our copyright guidelines: http://journals.plos.org/plosone/s/licenses-and-copyright.

  a. You may seek permission from the original copyright holder of Figures 1, 2, 3 and 5 to publish the content specifically under the CC BY 4.0 license.

Reviewers' comments:

Reviewer's Responses to Questions

**Comments to the Author**

1. Is the manuscript technically sound, and do the data support the conclusions?

Reviewer #1: Yes

Reviewer #2: Partly

2. Has the statistical analysis been performed appropriately and rigorously? 

Reviewer #1: Yes

Reviewer #2: No

3. Have the authors made all data underlying the findings in their manuscript fully available?

Reviewer #1: Yes

Reviewer #2: Yes

4. Is the manuscript presented in an intelligible fashion and written in standard English?

Reviewer #1: Yes

Reviewer #2: Yes

5. Review Comments to the Author

Reviewer #1: This paper proposes a new method for analysing landscape preferences using eye-tracking simulation software. Based on 78 panoramic landscape photographs, representing major landscape types of the Central European Alps, this study collected 19 indicators of the hotspots that were identified by the Visual Attention Software by 3M. Therefore, the method greatly saves manpower and resources, but there are a few places that the authors could improve upon.

In the introduction, you need to connect the evaluation of the landscape to your paper goals, such as natural landscape and mountain landscape. Please follow the literature review by a clear and concise state of the art analysis.

In the conclusion, in addition to summarizing the actions taken and results, please strengthen the explanation of their significance. It is recommended to use quantitative reasoning comparing with other types of landscape, especially those stemming from previous work.

Reviewer #2: The study tries to integrate eye-tracking simulation software into research on landscape preferences. Thereby the authors focus on the type of information that can be obtained through eye-tracking simulation and how this information can contribute to explain landscape preferences. In the end of the manuscript they conclude that you cannot declare landscape preferences this way.

The study convinces by a substantial combination of different methods and data. Everything is methodically collected cleanly and well brought together. But there is also some room for improvement.

Thus, the theoretical and methodological classification of the method should already start more clearly in the theoretical framework of this manuscript, including bottom up and top down processes in visual attention and locating the method of the 3M-VAS on initial attention at an early stage. A closer approximation to eye tracking methodology should also be sought in the method. Both orientations (methodological and theoretical) will help to remedy the weaknesses of the results and their interpretations.

6. PLOS authors have the option to publish the peer review history of their article (what does this mean?). If published, this will include your full peer review and any attached files.

Reviewer #1: No

Reviewer #2: No

---

## [Author Response · Author response to Decision Letter 0]

30 Jul 2022

Dear Editor, dear Reviewers,

we thank you for reviewing our manuscript and highly appreciate your time and effort. We have the impression that addressing your various constructive comments improved the manuscript significantly. 

Based on your suggestions, our main changes include

(1) a revision of the methods section by expanding the theory part especially with regard to top down and bottom up processes of visual attention and by adding more details to the different analysis steps;

(2) a recalculation of the regression models to avoid overfitting and an additional analysis of formerly unused outputs of 3M-VAS, explaining the contribution of different visual elements to the hotspots; 

(3) an improved framing of the manuscript by substantiating the introduction to better relate the evaluation of the landscape to our research objectives and by strengthening the discussion section to indicate more clearly the contribution of our findings.

In the following, we provide our detailed responses to each comment. 

Again, we thank you for your time and look forward to a positive outcome. 

Sincerely yours,

Uta Schirpke and co-authors

 

Reviewer #1: 

This paper proposes a new method for analysing landscape preferences using eye-tracking simulation software. Based on 78 panoramic landscape photographs, representing major landscape types of the Central European Alps, this study collected 19 indicators of the hotspots that were identified by the Visual Attention Software by 3M. Therefore, the method greatly saves manpower and resources, but there are a few places that the authors could improve upon.

In the introduction, you need to connect the evaluation of the landscape to your paper goals, such as natural landscape and mountain landscape. Please follow the literature review by a clear and concise state of the art analysis.

Response: Thank you. We revised the introduction, in particular, the first paragraph by adding information on mountain landscapes, their specific socio-ecological characteristics, landscape dynamics and impacts on aesthetic landscape values.

In the conclusion, in addition to summarizing the actions taken and results, please strengthen the explanation of their significance. It is recommended to use quantitative reasoning comparing with other types of landscape, especially those stemming from previous work.

Response: We substantiated this section by adding further details and more concrete examples to indicate more clearly the contribution of our findings to research and practice.

Detailed comments, questions and suggestions 

7, 141-143 In order to get the same range of the scale the data from one preference scale was rescaled from a 5- to a 10- point scale. 

Did you check for consistencies and did you compare the distributions of the measurements? There might be differences for the measurements that are based simply on the number of scale points. Especially reliability might be an issue regarding measurements with different numbers of scale points (see e.g. Weathers, Sharma, & Niedrich, 2005, https://doi.org/10.1016/ j.jbusres.2004.08.002 or Weijters, Cabooter, & Schillewaert, 2010, https://doi.org/10. 1016/j.ijresmar.2010.02.004). Especially since the data comes from different surveys, reliability might be one of the major problems. 

How did you cope with this issue? What did you do? Did you perform any additional analysis? If so, please report on it. 

Response: The three questionnaires were designed and structured in a similar way to reduce limitations in the comparability of the results (i.e., same picture format, same rating type, same sampling approach, same study area, etc.). Moreover, identical photos were included in each survey to standardize the survey results based on 16 photos. We have now revised this section to describe more clearly the methodological approach and the alignment of the preference scores.

7, 143-145 The surveys were carried out between 2011 and 2019, including residents and visitors, accounting for demographics and origin. 

Did you control for the influence of origin and sociodemographics? 

Response: Previous analyses of the three surveys demonstrated that that the differences between socio-demographic/cultural groups were small. In this study, we therefore calculated mean preference scores without differentiating between different groups. We added an explanation to the main text.

7, 149-150 Since the preferences of the landscapes serve as dependent variables the selection of the stimuli is central for the results of your study. Besides the selection criteria described in the text, how did you cope with a higher number of stimuli for one LULC type if all other criteria were fulfilled? Which pictures did you choose? Were the stimuli random or systematically chosen? Please add some more detail in this paragraph. 

Response:

The selection of the photographs for each LULC type followed a stratified sampling approach to cover as much of the spatial diversity of each LULC type as possible based on (1) different land-use intensity, (2) topography (with special attention to the slope) and (3) landscape context. We have revised the text accordingly.

7, 155 Please change ‘Eye tracking analysis’ into ‘eye tracking simulation’ analysis. 

Response: Done.

7, 156-157 Are there any known quality standards for 3M-VAS or measures of instrument reliability or validity? Please add some more information. 

Response: 3M-VAS does not provide much information about quality standards, but there is a limit in ‘nominal’ resolution under which the system will warn that it is insufficient (this was not the case in any of the images used here). All the system sees is the number of pixels and if one has up-scaled a low resolution image, it can fool it, but of course this is not helpful as no real resolution will have been added. We now added some information on this issue in the manuscript.

7-8,157-158 Was there a standardized upload of the images? Were the images comparable in size and resolution? Please provide further information. 

Response: The images were standardized for the questionnaires and not altered for uploading them in 3M-VAS, i.e., all images had the same size (2300x360 pixel) and resolution (300 dpi). The output images of 3M-VAS had all a size of 1024x160 pixel and a resolution of 96 dpi. We added this information in the manuscript.

8, 164-165 What do the percentages in the image describe? Is this the probability of the observation? Please add more details. 

What kind of data are these heatmaps based on? Is it comparable to time to first fixation, number of fixations, fixation duration? Please add more information in the caption to understand the figure. 

Response: The percentages describe the probability that areas are seen within the first 3-5 seconds. The heatmap is based on different visual elements (edges, intensity, red/green color contrast, blue/yellow color contrast, faces). To clearly present the outputs of 3M-VAS, we now included all available information and revised the figure caption to provide full details.

9, 175-176 Eye-tracking data are often considered dependent. This means, for example, that the emergence of a ‘hotspot’ through prolonged observation also minimizes the emergence of other hotspots. What about simulated data? Shouldn't they also be considered dependent on each other? And does this raise a problem for the analysis with methods for independent data? 

Response: The software treats each image regardless of others - unless of course they are presented in pairs - and results can be considered as independent. Whatever limitations may be a common denominator within all photographs, so they should not influence comparative results. We added a note on this issue in the manuscript.

10, 182-188 The proportion of clearly recognizable human structures, proportion of the sky, cloud cover and areas with special lighting conditions were estimated. Who estimated it? Were there coders? Were they trained, coached? According to which rules was the estimate made? Is there evidence of sufficiently high reliability and time stability of this measurements/estimations? Please add more information about this estimation process. 

Response: All estimations were performed by one of the authors to avoid uncertainty in the area estimates by different persons. This person first gridded some photographs for own calibration and then derived the proportions of certain features by the number of grids. This approach was discussed and determined in advance by the author team. We now added this information to the manuscript.

10, 192 

11, 210 Do you mean landscape metrics were obtained from study with reference number 10? would you be so kind and could rephrase the sentence into a more readable version. 

Response: We revised the sentences to improve readability.

11, 199-203 The idea behind the use of geodata is that you have the most accurate data possible about what you see in the images. 

But doesn't that assume a little too much knowledge about what you're seeing? The authors write that only visible areas are used. But what does visible mean? Does it mean that it is recognizable by color highlighting? Or does it have to be (a) clearly visually separable/distinguishable from the surrounding area as an independent area and (b) clearly and uniquely recognizable as a specific type of area (see also table 2)? Is a berry plantation also recognizable as such in the picture? Or is it simply a separately recognizable area without knowing what exactly it is? Do the authors add information based on the method that is not available to anyone who views the panoramic images? 

Please provide further information to justify the use of geodata and support the benefit of using it. 

Response: The distinguishability/recognisability of landscape features or LULC types depends on the distance from the observer. The geodata was therefore specifically prepared for each distance zones using different spatial and thematic resolution (see Table S1) to account for scale and perceived colour dependencies from distance. This means that, for example, the berry plantation would be a specific habitat type in zones 1 (at the observer point) and 2 (near zone), while it would not be distinguishable in zones 3 (middle zone) and 4 (far zone), i.e., it is merged with other land cover types, presenting a separately recognizable area without knowing what exactly it is. 

The visible area refers only to the area that can be seen by an observer in the landscape, as some parts are usually hidden by mountains. We therefore applied viewshed analysis to identify the area that is depicted on the photos also on the map, which we then used for calculating landscape metrics. We revised the methods section to explain more clearly the use of the distance zones as well as to improve the explanations on the visible area.

12 228-229 Thank you for creating the summary for all variables added to the regression. That helps to understand your model approach. 

Combining this information with the information about the number of cases (p7, lines 149-150), you are using 78 cases/observations (pictures) in a regression with (1) 8, (2) 19, (3) 30 and in the fourth regression (4) 49 regressors. 

Unfortunately, regression models with a high number of independent variables and a limited number of observations are prone to overfitting. Since all models have a rather unfavorable ratio of observations and explanatory variables, the problem seems to be relevant for your regression models. How did you cope with this issue? Since the sample and the complexity of the models can hardly be changed now, a clear limitation of the findings must be made in the discussion section. Maybe you are also able to reduce the complexity of the regression approach by including similar or opposite indicators (e.g., the HPmin HPmax values) in the model exclusively rather than together. This decision could be justified theoretically and statistically. 

To proof your model, you could also add more pictures or change the pictures and corresponding data in your regression. Approximately equal estimates show a reliable model approach. Did you change the pictures/their data and tested the regression models with more or other pictures/observations? What results did you achieve? This manuscript needs more evidence of your approach at this critical point. Add more information, also for cross validation of your models! 

Response: Thanks for the comment. We have taken care of various necessary preconditions for a validity of the results resulting from the linear regression, but actually did not consider the overfitting. Following this comment, we have now redone all model calculations using a 2-step procedure, which we have now also described in the method section. Due to the slightly altered results, we have also updated the results section.

18, 323-327 But there was the opportunity to gather some of these data about single elements using the unused reports from 3M-VAS (see p8, lines 161-162). Do you think there is more potential for using the additional outputs of the software? Is the analysis of this data too complicated to use? What was your experience? 

Couldn't this information have been collected via content analysis? 

Response: Thank you for this comment. We have taken up your suggestion and used the additional outputs of 3M-VAS to estimate the contribution of the different visual elements to the hotspots (edges, intensity, red/green color contrast, blue/yellow color contrast, faces). These elements are used by the software to identify the heatmap/hotspots, and this new analysis indeed provides deeper insights into bottom-up factors. Accordingly, we revised the description of the methods section and added the additional outputs to Fig. 3. We present the new findings in the results section and shortly mention them in the discussion section.

18, 331-332 One ‘)’ is missing somewhere. 

Response: Corrected.

18, 327-336 Thank you for this clear and quite concise summary of the weaknesses of the 3M-VAS method used. Unfortunately, the combination of (a) the limitation to "first glance" analysis combined with (b) the intentional exclusion of particularly salient visual stimuli such as lighting, shadows, etc. is a factor that severely minimizes the range of your results and maximizes the limitations of your study. 

In eye-tracking experiments on initial visual attention, the effects of salient stimuli are often tested. By minimizing salient stimuli, a break has arisen not only methodologically, but especially theoretically. 

This is expressed by the fact that it is not possible to say exactly which situation is now 'imitated' using the 3M-VAS data. It is not clear which (real) situation can be represented and analyzed with the simulated data. 

In order to draw a clearer picture of the scope of the findings, the discussion should be intensified. 

Also in context -lines 337-344 you should consider expanding the theory part of the paper a bit especially with regard to top down and bottom up processes of visual attention. This could also provide a clearer picture in the results and discussion sections. This would also contribute to a clearer classification of the findings 

Response: We have extended the description of the conceptual background by introducing the theoretical part on top down and bottom up processes of visual attention. In addition, we added more explanations to the discussion section to discuss our findings more clearly. 

20, 388-389 Regarding the mentioned points above it remains unclear if the evaluation of different scenarios by eye-tracking simulation really could provide important information for decision-making and landscape planning. In which planning or decision-making situation can the simulation of initial attention really alone or in combination provide sufficient evidence for highly complex decisions that are based not only on visual markers but also on economic, ecological or even other societal concerns on a meso or macro level? 

Please provide less abstract and more concrete ideas for using the findings of this study in real-world decision-making situations. 

Response: We revised this part of the discussion section by concretizing the examples for potential application of eye-tracking simulation. 

20, 390-397 Just like hearing, the combination of smelling and seeing or even feeling and seeing leads to a different perception. This perspective is of a more general nature and does not contribute anything to this specific study. Therefore, I suggest shortening or omit this discussion thread. 

Response: We agree and removed this paragraph from the manuscript.

 

Reviewer #2: 

The study tries to integrate eye-tracking simulation software into research on landscape preferences. Thereby the authors focus on the type of information that can be obtained through eye-tracking simulation and how this information can contribute to explain landscape preferences. In the end of the manuscript they conclude that you cannot declare landscape preferences this way.

The study convinces by a substantial combination of different methods and data. Everything is methodically collected cleanly and well brought together. But there is also some room for improvement.

Thus, the theoretical and methodological classification of the method should already start more clearly in the theoretical framework of this manuscript, including bottom up and top down processes in visual attention and locating the method of the 3M-VAS on initial attention at an early stage. A closer approximation to eye tracking methodology should also be sought in the method. Both orientations (methodological and theoretical) will help to remedy the weaknesses of the results and their interpretations.

Response: Thank you. We have extended the description of the conceptual background by introducing the theoretical part on top down and bottom up processes of visual attention. In addition, we added more explanations to the discussion section to discuss our findings more clearly.

---

## [Editor Report · Decision Letter 1]

10 Aug 2022

Potential of eye-tracking simulation software for analyzing landscape preferences

PONE-D-22-12355R1

Dear Dr. Uta Schirpke,

We’re pleased to inform you that your manuscript has been judged scientifically suitable for publication and will be formally accepted for publication once it meets all outstanding technical requirements.

Kind regards,

Ji-Zhong Wan

Academic Editor

PLOS ONE
---

## [Editor Report · Acceptance letter]

24 Aug 2022

PONE-D-22-12355R1 

Potential of eye-tracking simulation software for analyzing landscape preferences 

Dear Dr. Schirpke:

I'm pleased to inform you that your manuscript has been deemed suitable for publication in PLOS ONE. Congratulations! Your manuscript is now with our production department. 

Kind regards, 

on behalf of

Dr. Ji-Zhong Wan 

Academic Editor

PLOS ONE